# Evaluation of Salt Stress-Induced Changes in Polyamine, Amino Acid, and Phytoalexin Profiles in Mature Fruits of Grapevine Cultivars Grown in Tunisian Oases

**DOI:** 10.3390/plants12234031

**Published:** 2023-11-30

**Authors:** Abir Habib, Sihem Ben Maachia, Ahmed Namsi, Mounira Harbi Ben Slimane, Philippe Jeandet, Aziz Aziz

**Affiliations:** 1Horticulture Laboratory, Research Center of Oasis Agriculture (CRRAO), Deguech 2260, Tunisia; habibaboura87@gmail.com (A.H.); maachiasihem@yahoo.fr (S.B.M.); ahmed_en@yahoo.fr (A.N.); 2Horticulture Laboratory, National Agricultural Research Institute of Tunisia, Tunis 2049, Tunisia; mounira.bsh@gmail.com; 3Institut Supérieur Agronomique de Chott-Mariem, University of Sousse, Sousse 4000, Tunisia; 4RIBP USC INRAE 1488, Faculty of Sciences, University of Reims Champagne Ardenne, 51100 Reims, France; philippe.jeandet@univ-reims.fr

**Keywords:** amino acids, berry, grapevine, oasis table cultivars, phytoalexins, polyamines, high salinity, salt stress

## Abstract

Salinity stress has become an increasing threat to viticulture in the Tunisian oasis, and more generally, the characterization of salinity tolerance markers can be of great interest for sustainable grape production. This study investigated some metabolic adaptations in different tissues of the ripe berries of indigenous grapevine cultivars after exposure to salt stress to identify the key traits of salt stress tolerance under oasis conditions. We especially focused on the adaptive responses occurring at the level of amino acids, polyamines, and stilbene phytoalexins in the grape berry skin, pulp, and seeds of six grapevine cultivars differing in phenotypic and ampelographic characteristics. Our data showed that amino acids accumulated strongly in the pulp and skin, while resveratrol, trans-piceid and trans-ε-viniferin, as major phytoalexins, significantly accumulated in the seeds. High salinity was also found to increase both the berry skin and pulp contents of essential amino acids such as threonine, valine, leucine, isoleucine, lysine, methionine, and phenylalanine. The amounts of stilbenes also increased under high salinity in the berry skin of all the studied cultivars. Polyamine homeostasis within the different berry tissues suggested enhanced polyamine biosynthesis rather than polyamine oxidation in response to high salinity. Our principal component analysis revealed a clear discrimination between the cultivars based on their metabolic profiles within the ripe berry tissues under high salinity.

## 1. Introduction

Soil salinity is one of the major widespread challenges that severely threaten crop productivity worldwide. For the most important crops, yield losses are considerable and are mainly due to drought and high soil salinity, which are worsening in many regions due to global climate change. Salinity is known to limit water uptake and induce osmotic stress [1,2,3], as well as the accumulation of ions, leading to ionic stress [4,5,6] and nutritional disorders in plants [6,7].

Grapevine (*Vitis vinifera* L.) is economically the most important fruit crop in the world. In 2021, grape production was the world’s fourth largest type of fruit production, with 73.53 million metric tons of grapes being produced. Berries are eaten fresh as table grapes or industrialized for making juice, grape seed extracts, etc. [8]. Berry seed and skin extracts are also used for cosmetic and nutraceutical applications due to the pharmaceutical properties of some of their secondary metabolites, such as polyphenols and stilbenes [9,10]. Mediterranean countries are among the biggest producers of grapes [11], but recent climate changes, including high salinity in semi-arid areas, have become a huge threat to grape productivity. Although most grapevine cultivars are known to be moderately tolerant to soil salinity, their growth rate and physiological and metabolic status can be globally disturbed, thereby resulting in substantial losses in the yield and organoleptic properties of grapes and derived products [3]. Breeding programs have been developed regarding grapevine tolerance to abiotic stresses, and many cultivars of interest have been characterized [12]. It has been reported that more than 2000 genes are expressed in salt- and water-stressed Cabernet Sauvignon cultivars, and a large scale of expressed tag sequences have been found in Pinot Noir cultivars, which will be useful for genetic engineering strategies aimed at improving tolerance to biotic/abiotic stresses in grapevine [13,14]. Nevertheless, the mechanisms underlying salt tolerance in grapevine and especially in ripe berries remain very poorly understood. Therefore, characterizing salt-tolerant cultivars and elucidating the mechanisms involved in salt tolerance are of great importance in semi-arid areas [4].

Grape berries are non-climacteric fruits considered as sinks of primary and secondary metabolites essential for plant survival and development [15]. Depending on cultivar and environmental factors, berries undergo complex metabolic and biochemical changes during ripening [16,17]. The most notable changes concern the primary metabolism characterized by a rapid increase in both sugar and the major amino acid levels, as well as a progressive decrease in organic acids, during grape berry ripening [16,18]. Significant changes are also observed in the amounts of secondary metabolites, including the accumulation of polyamines and stilbene phytoalexins, as well as polyphenols and especially anthocyanins [19,20,21]. Overall, these compounds play an important role not only in determining the organoleptic quality and nutritional value of mature berries [21,22] but also in determining abiotic stress tolerance [15,17]. Some amino acids, especially those linked to glutamate metabolism, are known to play a major role in osmotic adjustment during salt stress [23,24]. They can also protect macromolecules and sub-cellular structures and mitigate the oxidative damage caused by reactive oxygen species produced in response to salt stress [25,26]. Polyamines also have a key role in regulating stress tolerance [23]. In combination with certain amino acids, such as proline and GABA, polyamines may indirectly contribute to osmotic adjustment by interacting with amino acid metabolism or hormonal signaling pathways [23,27,28]. More recently, we have shown that osmotic stress induces an increase in polyamine titers through reducing their oxidative degradation in grape berries, resulting in a lower accumulation of stilbene phytoalexins as a part of the grape defense mechanisms [17,29].

Various table grape cultivars differing in their phenotypic and ampelographic characteristics were prospected in the oases of southern Tunisia, where water resources are loaded with salt from 1.77 g·L^−1^ (electrical conductivity, ECe = 2.53 dS m^−1^) in conventional water resources to more than 5 g/L (ECe = 7.14 dS m^−1^) in surface well resources [30]. Such high salinity can lead to significant losses in grapevine productivity and changes in fruit composition, particularly with the semi-arid climates of these oases [31,32]. On the other hand, it has been suggested that the quality, especially the sweetness of fruits exposed to salt and/or drought stress during their production cycle, is improved by vegetative growth limitations [33]. The availability of such grapevine cultivars in the Tunisian oases offers an excellent opportunity to ensure the suitability of some cultivars to saline conditions or to select salt-tolerant grape cultivars for genetic improvement. Although the effects of salt stress on some metabolic profiles have already been described in leaves and ripe berries, the effects of salt stress on tissue-specific metabolic changes in ripe berries are still unknown.

Therefore, the aim of this study was to evaluate the impact of salt stress on the metabolic adaptation of mature berries from five native grapevine cultivars compared to an international cultivar grown in Tunisian oases. We particularly focused on the adaptive responses occurring at the level of amino acids as primary metabolites, and free polyamines and stilbene phytoalexins as secondary metabolites in the grape berry skin, pulp, and seeds in order to understand the relationships between the metabolic changes in different compartments of the ripe berry and the level of salt tolerance of native grapevine cultivars. The metabolic profiling of ripe berries revealed tissue-specific changes in response to salt stress depending on the susceptibility of the grapevine cultivar. This study could help others to refine and gain greater insights into the salt-induced metabolic changes in grape tissues and to identify key markers of salt stress tolerance among grapevine cultivars under oasis conditions.

## 2. Results

### 2.1. Polyamines in Berry Skin

Polyamines are strongly involved in osmotic adaptation as signal transduction molecules or solutes contributing to osmotic adjustment. An analysis of polyamines was carried out to understand the relationship between their profile and their accumulation at the berry tissue level and tolerance to salt stress. The analyses performed on the skin of the berries showed distinct polyamine profiles according to the grape cultivars (Figure 1). Overall, the main polyamines found in the berry skin were putrescine (Put) (about 50%), 1,3-diaminopropane (Dap, an oxidation product of polyamines), cadaverine (Cad, a product of lysine decarboxylation), the triamine spermidine (Spd), and the tetramine spermine (Spm) in small quantities. The berry skin of the cultivars Arbi, Chetoui, and Guelb Sardouk produced more polyamines compared to that of the cultivars Kahla, Muscat, and Sfaxi.

The effects of salt stress are also variable according to the cultivar. Salt stress induced a drastic reduction in polyamine levels in Arbi and Chetoui. However, an increase in polyamine contents was observed in Guelb S (Dap, Put, Spm) and Sfaxi (Dap, Cad, Spm), while only a slight impact of salt stress was observed in Kahla and Muscat regarding their polyamine contents in the berry skin. Nevertheless, Muscat accumulated a slight amount of Dap, accompanied by a reduction in Spd and Spm contents.

### 2.2. Amino Acids in Berry Skin

Knowledge of the role of amino acid metabolism in osmotic adjustment under saline conditions and plant tolerance mechanisms is still fragmentary. In order to determine whether the amino acid profile in the berry tissues correlates with the level of salt tolerance, analyses were carried out in the berry skin, pulp, and seeds of the grape cultivars. In the berry skin, amino acid analyses showed distinct profiles among the six grapevine cultivars (Figure 2, Appendix A). The most common amino acids found in berry skin are Gly, Arg, Glu, GABA, and Pro. Some cultivars, such as Arbi, Chetoui, and Muscat, displayed high amounts of amino acids in their berry skin. Guelb S and Sfaxi produced intermediate quantities, while Kahla produced smaller quantities.

Salt stress strongly affects the contents of amino acids, especially those involved in the metabolism of polyamines (arginine and ornithine), glutamate (Glu, Pro, GABA), and phenylpropanoids (Phe). Overall, in response to high salinity, there was an increase in the level of Arg in the berry skin of most cultivars, while Muscat and Chetoui showed a rather significant decrease in Arg content. Only Chetoui accumulated a high amount of Pro in response to high salinity, while the level of this amino acid decreased or did not change in the other cultivars. In the berry skin of Arbi and Chetoui, the content of Glu and GABA also increased, while both amino acids drastically declined in the berry skin of Sfaxi.

### 2.3. Stilbenes in Berry Skin

The importance of the phenylpropanoid pathway, and particularly of stilbene phytoalexin synthesis, in the stress response was clearly demonstrated. However, the role of stilbenes in the tissue response of ripe berries to salt stress is still unknown. In this study, the main stilbenes (monomeric and dimeric forms) were analyzed to determine whether high salinity could affect their contents in the berry tissues of the table grape cultivars. Resveratrol, its glycosylated form piceid, the resveratrol oligomer ε-viniferin, and its isomer δ-viniferin were identified in the berry skin of the six table grape cultivars, but with different accumulation profiles (Figure 3). Piceid was the most abundant stilbene in most cultivars, while ε-viniferin was not produced in the berry skin of the tested grapevines under the control conditions. Chetoui displayed high resveratrol levels under the control conditions, while the other cultivars produced smaller amounts.

The effects of salt stress were variable depending on the cultivar and the type of stilbenes. Piceid levels declined in the berry skins of the studied table grape cultivars, except for that of Kahla, which accumulated significant piceid amounts under high salinity. Trans-resveratrol levels significantly increased in the berry skin for all cultivars in response to salinity stress. This increase was more important for Chetoui, Muscat d’Italie, and Sfaxi. Among the resveratrol dimers, trans ε-viniferin strongly accumulated only in the berry skin of the tolerant cultivar Kahla under salinity stress. A similar trend was observed regarding the trans-δ-viniferin level in the berry skin of the tolerant cultivars Arbi, Kahla, and Chetoui under high-salinity conditions, with this compound being absent in the less tolerant ones, Sfaxi and Muscat d’Italie.

### 2.4. PCA Plot and Heatmap for Berry Skin Metabolites

The two principal components (Figure 4A) accounting for 55.42% of the total variance illustrate significant differences between the control and high-salinity conditions regarding the levels of metabolites in the berry skin. The cultivars Arbi (A) and Guelb S (G) were clearly discriminated from the other cultivars, with a higher level of polyamines (Spm and Dap) and amino acids (Gln, Arg, Gly, Val, Tyr, Lys, Orn, Aspn, Cys, Asp, and Cystin) under high salinity. However, under the control conditions, Guelb S (G) and Sfaxi (S) were distinguished by their high contents of Glu, GABA, Ala, Ser, Aspn, Phe, Leu, Ile, and Met.

The heatmap for the metabolic composition of the berry skin (Figure 4B) showed the same discrimination, revealing two groups of cultivars. The first one includes Arbi (A) and Guelb S (G) grown under high salinity and Sfaxi (S) and Guelb S (G) grown under the control conditions (Ctrl), with high levels of amino acids being present in the berry skins. The second group showed the lowest levels of Orn, Lys, Tyr, and Val (Figure 4B). Proline sharply accumulated in the berry skin of Chetoui (C) in response to high salinity. Regarding the phytoalexin response, Sfaxi (S), Chetoui (C), and Muscat d’Italie (M) were characterized by an accumulation of resveratrol in the berry skin in response to high salinity, while piceid, ε-viniferin, and δ-viniferin were highly accumulated in the berry skin of Kahla (K).

### 2.5. Polyamines in Berry Pulp

The analyses carried out on the berry pulp of the studied table grapes revealed different amounts of polyamines (Figure 5). The most abundant polyamine was Put, followed by Spd, Dap, and Cad, while Spm was found in small amounts. Under the control conditions, most of the table grape cultivars had significant amounts of polyamines in their berry pulp. Chetoui produced more polyamines than the other cultivars. Kahla, Guelb S, Muscat d’Italie, and Sfaxi accumulated intermediate and similar quantities of polyamines, while Arbi produced smaller amounts of Put, Cad, Spd, and Spm. Exposure to high salinity resulted in enhanced levels of Put, Spd, and Spm in berry pulp, especially in Chetoui and Muscat d’Italie. A significant increase in polyamine content was also observed in the berry pulp of Guelb S and Sfaxi; a slight increase in polyamine content was also observed in the berry pulp of Arbi. Cadaverine also accumulated in the pulp of Arbi, Chetoui, and Sfaxi. However, the polyamine level decreased in the berry pulp of Kahla.

### 2.6. Amino Acids in Berry Pulp

Amino acid profiling in berry pulp showed distinct amounts of these compounds among grape cultivars (Figure 6, Appendix A). Under control conditions, Chetoui and Guelb S produced lower amounts of amino acids in their berry pulp compared to the other cultivars. Herein, the high salinity increased the berry pulp amino acid contents, especially the contents of those involved in polyamine metabolism (Arginine and ornithine) and that of phenylpropanoids (Phe). Guelb S also showed an increase in proline and GABA in its berry pulp; Kahla and Muscat d’Italie accumulated GABA in their berry pulp. However, high salinity induced a significant accumulation of proline but reduced the level of arginine in the berry pulp of Chetoui. The other amino acids did not show any significant changes under salt stress.

### 2.7. PCA Plot and Heatmap for Berry Pulp Metabolites

Our PCA analysis defined two groups based on the metabolic composition of the berry pulp of the grape cultivars under salt stress conditions (Figure 7A). Arbi and Guelb S belonged to the first group under high salinity. Chetoui at high salinity was well discriminated from the other cultivars. Our PCA analysis revealed that the first and second axes explained 44.80% and 14.62% of the total variance; the two principal components accounting for 59.41% of the total variance (Figure 7A).

The first axis was highly correlated with the amino acids Arg, Cys, Tyr, Ile, Gly, and Aspn, while the second was highly correlated with the polyamines Dap and Put (Figure 7A). Furthermore, the constructed heatmap (Figure 7B) supported the discrimination of the tested cultivars based on the metabolic composition of their berry pulp. This showed that the first group, represented by Arbi and Guelb S, displayed the highest levels of almost all amino acids, including the essential amino acids Gly, Arg, Asp, and Gln (Figure 7B). Sfaxi, under the two growing conditions, and Muscat d’Italie were grouped due to the accumulation of Glu, Ala, Ser, Gly Arg, and Thr. The third group showed the lowest levels of almost all amino acids. The lowest levels of Leu, Ile, Met, Thr, Phe, and Aspn were recorded in the berry pulp of Arbi, Chetoui, and Guelb S under low-salinity conditions. Muscat d’Italie and Kahla demonstrated growth under the control conditions; Chetoui and Kahla demonstrated growth at high salinity and showed the lowest levels of Gln, Tyr, Cys, Orn, Gly, Arg, Lys, Asp, and Val. However, at high salinity, Chetoui accumulated the highest levels of Pro, Dap, Spm, and Put (Figure 7B).

### 2.8. Polyamines in Berry Seeds

The table grape cultivars exhibited different polyamine profiles in their berry seeds (Figure 8). Dap, Put, Cad, and Spd were found in similar amounts; spermine was produced in smaller quantities. However, in Kahla and Sfaxi, Dap (an oxidation product of polyamines) was detected at higher levels compared to the other polyamines (Put, Cad, Spd, and Spm). The effects of salt stress varied depending on the cultivars. These effects resulted in a decrease in putrescine content and an increase in Spm and/or Spd contents. The Dap content in berry seeds increased in Arbi, Kahla, and Muscat d’Italie but decreased in Chetoui. Compared to the other cultivars, Sfaxi showed a significant accumulation of putrescine in berry seeds in response to high salinity, associated with reductions in the contents of Dap, Spd, and Spm.

### 2.9. Amino Acids in Berry Seeds

The analyses performed on berry seeds showed distinct amino acid profiles depending on the cultivar (Figure 9; Appendix A). Arbi and Chetoui exhibited a high amino acid content in berry seeds; Kahla, Guelb Sardouk, and Sfaxi had intermediate levels, while Muscat d’Italie produced smaller amounts of amino acids. The high salinity significantly affected the amino acid levels in the berry seeds of the studied table grape cultivars, with the exception of Kahla and Muscat d’Italie. Overall, the grape cultivars showed decreased levels of amino acids in their berry seeds, especially those involved in the metabolism of polyamines (arginine and ornithine), glutamate (Glu, Pro, GABA), and phenylpropanoids (Phe). Yet, Sfaxi and Guelb S exhibited proline accumulation in their berry seeds in response to high salinity. For Kahla and Muscat d’Italie, exposure to high salinity resulted in an increase in the amount of almost all amino acids in their berry seeds.

### 2.10. Stilbenes in Berry Seeds

Our stilbene analyses showed distinct patterns in berry seeds among cultivars (Figure 10). Unlike berry skin, piceid was an abundant stilbene in the berry seeds of Muscat d’Italie, but its content was very low in Arbi. The other v cultivars produced comparable amounts of piceid under control conditions. Importantly, the tolerant cultivars also accumulated higher amounts of resveratrol (Arbi, Kahla, and Chetoui) and ε-viniferin (Kahla and Chetoui) under control conditions. δ-viniferin was produced in small amounts in the berry seeds of Arbi and Sfaxi but was not detected in the other cultivars. The effects of salt stress globally resulted in a drastic reduction in the stilbene contents of the berry seeds of the table grape cultivars. Under high-salinity conditions, an overall decrease in trans-resveratrol levels was observed in the berry seeds for the tolerant cultivars (Arbi, Kahla, Chetoui, and Guelb S), while the trans-resveratrol levels remained stable in the less tolerant ones (Sfaxi and Muscat d’Italie). The piceid levels remained stable in Kahla and Sfaxi but decreased under high-salinity conditions in the other cultivars (Arbi, Chetoui, Guelb Sardouk, and Muscat d’Italie). Among the resveratrol dimers, the contents of trans ε-viniferin and trans-δ-viniferin decreased or remained stable in the berry seeds of almost all cultivars under salinity stress, although increases were observed for ε-viniferin and δ-viniferin in the tolerant cultivars (Arbi and Chetoui and Kahla, respectively).

### 2.11. PCA Plot and Heatmap for Berry Seed Metabolites

Our PCA analysis of the metabolic compositions of the berry seeds showed two groups according to the grape cultivars (Figure 11A). The grape cultivars of each group were well discriminated. Kahla, Sfaxi, and Muscat d’Italie (Ctrl and high salinity) belonged to the first group, while Arbi, Chetoui, and Guelb S (Ctrl and high salinity) belonged to the second group. As shown in Figure 11A, the first and the second axes explained 38.13% and 20.76% of the total variance; the two principal components accounted for 58.89% of the total variance. These findings were supported by the heatmaps of the various metabolites in the berry seeds (Figure 11B). Otherwise, the heatmaps revealed that the highest levels of resveratrol and ε-viniferin were recorded in Kahla under the control conditions. The amino acids Thr, Ser, Ala, Cys, Gln, Asp, Arg, Gly, Orn, Cystin, Tyr, Met, Pro, and GABA, as well as the stilbene piceid, strongly accumulated in the berry seeds of the first group, especially in Sfaxi under the control and salt stress conditions and in Muscat d’Italie under low-salinity conditions (Figure 11B). The levels of these amino acids, as well as those of piceid, were low for the cultivars of the second group. Interestingly, the amino acids Val, Aspn, Phe, Lys, and Glu; the polyamine Spm; and the phytoalexin δ-viniferin accumulated in the berry seeds of Arbi at low salinity. Chetoui exhibited elevated levels of Orn, Leu, Ile, Glu, Lys, and Aspn in its berry seeds at high salinity.

## 3. Discussion

High-salinity conditions have become a major threat to food production in semi-arid areas. Thus, understanding the mechanisms involved in the adaptation to salt stress is of great importance to improve plant tolerance. In this study, we examined the metabolic profiles of the amino acids, polyamines, and stilbenes of the berry components (skin, pulp, and seeds) of six grapevine cultivars differing in their phenotypic and ampelographic characteristics to assess their responses to high-salinity conditions. These cultivars showed large differences in phenotypic traits at the shoot level but also in berries and clusters at the ripening stage (e.g., growth, morphology, architecture, yield) (Appendix A). Such phenotypic traits may be relevant for selection purposes regarding abiotic stress tolerance. Analyzing the metabolic profiles of mature berries could also be of great importance in adaptive and phenotyping strategies. During ripening, berries are highly susceptible to environmental changes, and their response to these changes varies depending on the grape cultivar. Although some physiological and metabolic responses have been characterized in the berries of different grape cultivars, the contribution of the berry tissues to the salt stress response and their exploitation in phenotyping and breeding strategies remain unknown. In this study, comparative trials with six grapevine cultivars revealed that there is some variability in the targeted metabolic profiles (polyamines, amino acids, and stilbenes) at the tissue level of ripe berries and their adaptation to high-salinity conditions.

### 3.1. Polyamine Titers in Response to Salt Stress

Polyamines are low-molecular-weight organic cations that are known to play a prominent role in the adaptation of plants to abiotic stress [23,34]. Our data showed that a significant accumulation of putrescine, spermidine, and spermine took place in the pulp of all cultivars in response to high salinity (Figure 5). However, at the berry skin level, a significant reduction in polyamine contents was observed in Arbi (the most tolerant) and Chetoui (Figure 8). Guelb S and Sfaxi showed enhanced levels of polyamine titers in their skin, pulp, and seeds (Figure 1, Figure 5 and Figure 8). Only a slight impact of salt stress was observed in Kahla (moderately tolerant) and Muscat d’Italie (susceptible) regarding their polyamine contents in berry skin (Figure 1). Overall, these changes correlate with an increased or unchanged level of 1,3-Dap (an oxidation product of spermidine and spermine) in the berry tissues of the oasis grape cultivars in response to salt stress (Figure 1, Figure 5 and Figure 8). However, reductions in 1,3-Dap were observed in the berry pulp of Muscat d’Italie (susceptible) and the berry seeds of Chetoui and Sfaxi (Figure 5 and Figure 8). Such an effect could be linked to the activation of polyamine oxidation through diamine- and polyamine-oxidases, as reported in the berries of another grapevine cultivars, such as Chardonnay, in response to osmotic stress and abscisic acid (ABA) treatments [17]. Therefore, our PCA analyses indicated that the cultivars Arbi and Guelb S were clearly discriminated from the other cultivars, with higher levels of polyamines in their berry tissues (Figure 4, Figure 7 and Figure 11), suggesting that salinity tolerance in the oasis cultivars involves the activation of the biosynthesis of polyamines and their subsequent oxidation, thereby contributing to metabolic homeostasis [35,36]. These results are consistent with other previous works which suggest that maintaining a certain level of polyamine biosynthesis in grapevine tissues under salt stress conditions could contribute to increased ROS scavenging activity and the protection of the photosynthetic apparatus against oxidative damages [23,37]. Due to their polycationic character, polyamines can bind to negatively charged surfaces, thereby preventing membrane injury from salt-induced ionic stress [34] and restricting Na^+^ influx and salt-induced K^+^ efflux [38,39].

The catabolism of polyamines via diamine and polyamine oxidases also generates H_2_O_2_, which has been proposed to play an important role in abiotic stress signaling and adaptation [29,40,41,42,43]. 1,3-Dap could also be oxidized into 3-aminopropionaldehyde to generate the osmoprotectants β-alanine betaine and γ-aminobutyric acid (GABA), suggesting a role for polyamine oxidation in salt stress tolerance [44]. This is consistent with the observation that increased levels of polyamines in the berry pulp of Muscat d’Italie and in the berry seeds of Sfaxi are associated with a decrease in the amounts of 1,3-Dap (Figure 5 and Figure 8). Similarly, it has been shown that mature berries of Chardonnay (a susceptible cultivar) display increases in free polyamine levels after exposure to ABA, while the concentration of 1,3-Dap decreases significantly [17]. This imbalance between the activation of the biosynthetic and oxidative pathways can lead to an excess or lack of polyamines, conferring sensitivity to salt stress. However, the changes in the polyamine titers produced in the berry skin of Arbi (tolerant) and in the berry pulp of Kahla (mid-tolerant) could be due to enhanced polyamine synthesis and oxidation. Therefore, one could argue that the activated turnover of polyamines in the berry skin and pulp of tolerant cultivars could support metabolic homeostasis contributing to the salt stress tolerance process.

### 3.2. Amino Acid Patterns in Response to Salt Stress

The effects of salt stress on amino acid metabolism have been well documented and can be related to the activation of the biosynthesis of the amino acids involved in osmoregulation, to a direct consequence of the toxic effects of Na^+^, or to an imbalance between protein biosynthesis and degradation. The current study’s results reveal significant variability in the distribution of amino acids between the different berry components of the grapevine cultivars. The oasis grape cultivars displayed the lowest levels of amino acids in their berry seeds (Figure 9). The main differences between the cultivars concern the amino acid composition of the berry skin and pulp (Appendix A). A positive correlation was observed between salt tolerance (in the cases of Arbi, Chetoui, Guelb S, and Sfaxi) and the accumulation of amino acids in the berry skin and pulp (Figure 2 and Figure 6). Muscat d’Italie, as a susceptible cultivar, accumulated amino acids in its pulp and seeds (Figure 6 and Figure 9). The amino acids with a positive correlation with salt tolerance were those related to polyamine metabolism (arginine and ornithine), nitrogen assimilation and transport (Glu, Pro, GABA), and phenylpropanoid biosynthesis (Phe). The more tolerant cultivars (Arbi and Chetoui) showed higher levels of Glu and GABA in their berry skin, while both of these amino acids were decreased in the berry skin of Sfaxi (Figure 2). This increase in amino acid content could be attributed to the global activation of their biosynthetic pathways, initiated by reduced water potential during salt stress [33].

The most discriminating traits between the cultivars were the highest levels of almost all amino acids, including the essential amino acids Gly, Arg, Asp, and Gln for Arbi and Guelb S as tolerant cultivars, while Sfaxi and Musca d’Italie (susceptible) accumulated Glu, Ala, Ser, Gly, Arg, and Thr under low- and high-salinity conditions. Chetoui and Kahla, in response to high-salinity conditions, showed the lowest levels of Gln, Tyr, Cys, Orn, Gly, Arg, Lys, Asp, and Val. However, Chetoui accumulated the highest levels of Pro in response to high salinity (Figure 4, Figure 7 and Figure 11). These patterns could be related to a salt stress-induced disturbance in water potential between organs/tissues, leading to a decrease in turgor [45]. Our grape cultivars also exhibited a great accumulation of phenylalanine, especially within the pulp of their berries; phenylalanine is an aromatic amino acid that serves as a precursor for the synthesis of several secondary metabolites, such as stilbenes and flavonoids, under abiotic stress conditions [46].

The metabolism of amino acids in plants is closely linked to the cellular homeostasis of polyamines, particularly the metabolism of arginine and ornithine—the main precursors of the biosynthesis of putrescine—and that of GABA and Pro, which may result from the catabolism of Put [47]. It has been reported [29] that water stress tolerance in grapevine genotypes is associated with an improved metabolic flux of Arg to GABA and Pro. Interestingly, Pro accumulated in the berry tissues (skin, pulp, and seeds) of Chetoui in response to high salinity; its level also increased in the berry seeds of Sfaxi and in the berry seeds and pulp of Guelb S (Figure 2, Figure 6 and Figure 9). As for the non-protein amino acid GABA, it slightly accumulated in the berry tissues of the tolerant cultivars Arbi, Chetoui, Guelb Sardouk (berry skin), and Kahla (pulp and seeds). A portion of GABA can also serve as a useful metabolic substrate, providing energy to maintain higher membrane potential and provide carbon skeletons under stress [33,48].

### 3.3. The Accumulation of Stilbenes as a Part of Adaptation to Salt Stress

Stilbene accumulation is one of the most studied defense responses in grapevine due to their importance as phytoalexins exhibiting various beneficial activities [9,49,50]. Their biosynthesis is often modulated by many abiotic and environmental factors, the most studied of which include temperature, UV radiation, and drought stress [23,51,52]. More recently, it has been shown that osmotic stresses, including salinity, are also abiotic factors capable of modulating grapevine defense responses, including stilbene biosynthesis [29,52,53]. High levels of stilbene resveratrol production, and high levels of its glucosylated form piceid, were observed in the sensitive cultivar Syrah when exposed to salt stress, while Razegui, being salt tolerant, did not show any significant change towards these stilbenes [53]. Resveratrol is the iconic representative of a chemical group of more than 1000 compounds comprising glucosylated, isoprenylated, methylated, and hydroxylated derivatives, with the majority of this group being unmistakably represented by resveratrol oligomers [9,49,54]. The most studied and well-known resveratrol derivatives are piceid, the 3-O-β-D-resveratrol glucoside, and the two dimers δ and ε-viniferins. These three compounds, together with resveratrol, have been analyzed in this study as they can be considered representatives of the grapevine phytoalexin response.

The profiles of the four stilbenes analyzed in the present work (resveratrol, piceid, and the two dimers δ and ε-viniferins) were obtained from extracts of the berry skin and the seeds (the pulp does not produce stilbenes) of all the cultivars with the aim of searching for possible correlations between the production levels of these compounds and the cultivars’ supposed tolerance to salt stress, taking into account that the Arbi, Guelb S, Kahala, and Chetoui cultivars are considered tolerant and the Sfaxi and Muscat d’Italie cultivars are considered to be susceptible (Figure 3 and Figure 10). Our analysis of the data showed several trends: (1) Apart from δ-viniferin, the concentrations of resveratrol, piceid, and ε-viniferin in the berry seeds were higher than in the berry skin regardless of the cultivar and whether or not the cultivar was exposed to salt stress. (2) Resveratrol levels were consistently and significantly higher in the berry skin of all cultivars in response to high salinity unrelated to cultivar tolerance to salt stress. (3) The observed accumulation of the more antioxidant viniferins in the salt-stress-tolerant grape cultivars (Arbi, Chetoui and Kahla for δ-viniferin; Chetoui and Khala for ε-viniferin) is of paramount importance. A significant increase in the skin piceid content was also observed in response to salt stress in the tolerant cultivar Kahla. (4) Stilbene concentrations generally decreased or remained unchanged in the berry seeds of all cultivars after exposure to high-salinity conditions (ECe = 8.02 dS m^−1^). Surprisingly, the stilbene concentrations (monomers and dimers) in the berry seeds of all cultivars were higher than in the berry skin, and this finding is somewhat at odds with pioneering works on stilbene accumulation in grape berries that have evidenced that grape skin is the main tissue for stilbene biosynthesis [55,56]. The fact that the stilbene contents were higher in the seeds than in the skin where the production of stilbenes is localized suggests the transport of these compounds from the skin to the seeds. Some studies have indeed reported the possible transport of these compounds via the phloem route [57].

Interestingly, in this study, we also obsevred a significant accumulation of resveratrol in the berry skins of all cultivars in response to high-salinity conditions (Figure 3). Some works have reported better plant adaptation to salt stress in response to the exogenous application of phenolics, α-tocopherol, and resveratrol [51,58]. Indeed, resveratrol (100 µmol), used alone or in combination with 0.5 mmol α-tocopherol, was found to diminish the level of salt-induced oxidative stress through increases in antioxidant enzyme activities, the recovery of ion homeostasis, and the photosynthetic pigment content, resulting in the restoration of almost all photosynthetic parameters (stomatal conductivity and photosynthesis rate) in citrus [51] and apple seedlings (*Malus hupehensis* Rhed) [58]. These two works clearly illustrate the key role of phenolics and particularly of resveratrol in relieving plants from salt stress. The increases in the concentration of the dimers (δ-viniferin and ε-viniferin) and piceid reported in this study for the berry skin of the cultivars tolerant to salt stress, as well as the resveratrol contents of all cultivars in response to high-salinity conditions (Figure 3), further confirm the role of stilbenes in the adaptive response of plants to salt stress [51,53,58]. In the same way, a very recent work [4] reported a strong induction of stilbene synthase transcripts in a Tunisian wild grape cultivar as a consequence of salt stress, thus confirming the influence of the phytoalexin response in the mechanisms of plant tolerance to high salinity. The accumulation of dimers in tolerant cultivars is of particular interest as these compounds exhibit higher antioxidant activities than the monomers, possibly efficiently counteracting the effect of oxidative stress linked to high-salinity conditions.

## 4. Materials and Methods

### 4.1. Site Selection and Climate of the Oases

The oases of El Jerid occupy a small part of the Sahara in the governorate of Tozeur in southwestern Tunisia. They are located northwest of Chott El Jerid, with approximately 1096732 date palms. The oases are characterized by a harsh and arid climate, with an often mild and slightly rainy winter and a dry and hot summer. The recorded rainfall explains its belonging to the upper Saharan bioclimatic stage. This region receives very low annual rainfall (less than 100 mm per year) [59]. The average annual temperature exceeds 21 °C. Over the study period, the temperature of the hottest month (July) was 32.97 °C, and that of the coldest month (December) was 13.73 °C. The field experiment was conducted on twenty-year-old indigenous table cultivars of *Vitis vinifera* L. (Arbi, Chetoui, Guelb Sardouk, Kahla, and Sfaxi) and an introduced cultivar (the Muscat d’Italie grown in the oasis of El Jerid). Guelb Sardouk and Kahla are two red grape cultivars. These cultivars showed a good adaptation to the oasis conditions, particularly to the high-salinity load of the irrigation water [30]. The ampelographic characteristics of these grape cultivars are shown in the Appendix A.

Two experimental plots were chosen according to the salt loads of the irrigation water (Appendix A). Plot 1 was irrigated with conventional water (control) with an electrical conductivity (ECe) of 3.945 dS m^−1^ (salinity of 2.76 g·L^−1^). For plot 2, the water was supplied by the surface well with an electrical conductivity of 8.02 dS m^−1^ (salinity of 5.6 g·L^−1^). All cultivars were planted with a spacing of three meters between rows and 1.5 m between vines. The vines were trained on trellises oriented vertically in an East–West orientation. Throughout the period from veraison to ripening (June to September), the external temperature recorded in the oasis was relatively constant (between 31 and 32.97 °C). Over the same period, rainfall was very low and close to zero (0–0.6 mm). Irrigation was applied every four days based on the evaporation rate. To avoid any border effects, only the plants located in the middle of each row were used for the experiment. The experimental plot was bordered with a row of vine receiving a similar irrigation amount.

### 4.2. Sample Collection

Harvest was performed at optimum fruit maturity, when the fruits reached the desired degree Brix and when almost the entire surface of the berries reached a red color (for Guelb Sardouk and Kahla cultivars). Sampling was performed at full ripening (at the end of June for Arbi; during August for Guelb Sardouk, Kahla, and Muscat d’Italie; and during September for Sfaxi and Chetoui). Grape bunches were collected from 3 trees per cultivar for both experimental plots. The berry skin, pulp, and seeds were then separated from the ripe berries at room temperature and stored at −20 °C. The berry pulp tissues were lyophilized. Then, all tissues were ground to powders in liquid nitrogen and stored at −20 °C prior to analysis.

### 4.3. Amino Acid Analysis

Amino acid extraction was performed on powdered samples of the berry skin, pulp, and seeds as described in Hatmi et al. [23]. The powders (50 mg) were suspended in 400 µL of methanol (100%), followed by 15 min of stirring at room temperature. Then, 200 µL of chloroform was added, followed by 5 min of stirring. After that, 400 µL of ultrapure water (UPW) was added. The samples were vortexed vigorously to induce phase separation and then centrifuged for 5 min at 13,000× *g*. Aliquots of 200 µL were collected from the upper phase containing amino acids, transferred to clean vials, and dried under vacuum. Finally, the dry residues were suspended in 50 µL of UPW (vortex step), and 10 µL was used for derivatization using the AccQ^TM^-Tag Ultra derivatization kit (Waters Corporation, Milford, MA, USA) [23].

The derivatized amino acids were analyzed using an Acquity UPLC system (Waters) as described in Hatmi et al. [23]. Two µL of the reaction mixture was injected onto an Acquity^TM^ UPLC BEH C18 1.7 mm 2.1 × 100 mm column heated at 55 °C. Amino acids were eluted with a mixture of AccQ^TM^-Tag ultra-eluent A that had been diluted 20 times (Waters Corporation, Milford, MA, USA) and pure acetonitrile at a flow rate of 0.7 mL·min^−1^ according to the following gradient: initial, 99.9% A; 0.54 min, 99.9% A; 6.50 min, 90.9% A; 8.50 min, 78.8% A; 8.90 min, 40.4% A; 9.50 min, 40.4% A; 9.60 min, 99.9% A; and 10.10 min, 99.9% A. Derivatized amino acids were detected using an Acquity^TM^ fluorimeter (Waters Corporation, Milford, MA, USA) with an excitation wavelength of 260 nm and an emission wavelength of 473 nm. Amino acids were characterized via co-chromatography of individual standards and quantified after normalization against internal standards and plant material dry weight.

### 4.4. Polyamine Analysis

Free polyamines (PAs) were extracted from plant materials (freeze-dried powders of berry skin, pulp, and seeds) with cold 1 M HCl (0.05:1, *w*/*v*) on ice as described by Aziz et al. [60]. Dansylation was carried out by mixing 250 μL of the aliquots (from the supernatants) with 600 μL of dansyl chloride (10 mg·mL^−1^) and 100 mg of sodium carbonate. After adding the proline solution (300 μL of 100 mg·mL^−1^), the mixtures were dried under a stream of nitrogen to remove the excess dansyl chloride. The dansylated polyamines were further extracted using ethyl acetate. Then, the organic phase was dried under a stream of nitrogen. The residue was solubilized with 1 mL of methanol, filtered through 0.22 μm PTFE filters, and stored at −20 °C until analysis.

Dansyl-polyamines were analyzed using an Acquity^TM^ UPLC system (Waters Corporation, Milford, MA, USA). They were eluted on a BEH C18 1.7 µm 2.1 × 100 mm column heated at 30 °C with acetonitrile/water, with a solvent gradient changing from 50 to 95% over 5 min at a flow rate of 0.5 mL·min^−1^; elution was completed in 0.5 min, and the column was washed with 100% acetonitrile for 1 min. Dansyl-polyamines were detected using an Acquity^TM^ fluorometer (Waters Corporation, Milford, MA, USA) with an excitation wavelength of 365 nm and an emission wavelength of 510 nm. Polyamines were identified and quantified after calibration with external standards (Sigma).

### 4.5. Stilbene Analysis

Stilbene phytoalexins were extracted from freeze-dried powder tissues (berry skin, pulp, and seeds) with 1 mL of methanol: water 85% (*v*/*v*). Tubes were shacked at room temperature for 2 h then centrifuged for 10 min at 15,000× *g*. The supernatant was dried under a stream of nitrogen. The residues were recovered by 1 mL of methanol and filtered through 0.22 µm PTFE filters. Stilbenes were then analyzed according to Hatmi et al. [23], using an Acquity^TM^ UPLC system (Waters Corporation, Milford, MA, USA) with a gradient from 10 to 90% acetonitrile/water over 7 min at a flow rate of 0.5 mL·min^−1^. The column was an Acquity^TM^ UPLC BEH C18 1.7 µm 2.1 × 100 mm, heated at 30 °C. Stilbenes were detected with an Acquity^TM^ fluorometer (Waters) at an excitation wavelength of 330 nm with an emission wavelength of 375 nm. Stilbene phytoalexins were identified and quantified with reference to retention time and calibration with external standards.

### 4.6. Statistical Analyses

Three replicates were used for each condition. Data for the control and high-salinity conditions within each parameter were compared using an analysis of variance (ANOVA) followed by the Newman–Keuls test. Differences between the mean values were considered significant at *p* < 0.05. The principal component analysis (PCA) plots and the heat maps summarizing the metabolic composition of the different berry components (skin, pulp, and seeds) according to the salinity of the water irrigation were generated using Xlstat (2016) with Euclidean distance as a similarity measurement and hierarchical clustering with complete linkage. A false color scale was used to show the level of each metabolite in the control and salt-stressed cultivars. Green shades indicate overproduction, and red ones correspond to a lower amount of metabolites. Statistical analyses were performed using SPSS (IMB SPSS statistics 20) and Xlstat (2016).

## 5. Conclusions

This study has highlighted the importance of studying the metabolic patterns of amino acids, polyamines, and stilbenes at the level of mature berry tissues of grapevine cultivars cultivated under high-salinity conditions. The results revealed that high salinity induced various changes in the homeostasis of the studied metabolites in the berry skin, pulp, and seeds of the oasis table grape cultivars and also highlighted tissue-specific changes in response to salt stress. We have also shown that there is some variability in the metabolic profiles of berry tissues among grapevine cultivars regarding their tolerance to high-salinity conditions. The salt-tolerant cultivars Arbi and Guelb S were clearly discriminated from the other cultivars, with a higher level of polyamines in their berry tissues and a greater accumulation of amino acids, especially those related to polyamine and phenylpropanoid metabolism, in their berry skin and pulp. Our data also illustrate the contribution of an enhanced accumulation of stilbenes and resveratrol and its dimers (δ-viniferin and ε-viniferin) in berry skin in the grapevine cultivars’ adaptive responses and tolerance to salt stress. The information contained in this work thus provides in-depth insights into the grapevine metabolic adjustments taking place at the tissue level of ripe berries, and these insights could prove valuable in improving salt tolerance and exploring some metabolic traits which may represent useful markers for the selection of more suitable cultivars or genotypes, or for the construction of a phenotyping strategy that helps vineyards face the challenge of climate change. Studies on a broader set of grapevine cultivars are needed to unveil whether these metabolic responses, along with other phenotypic traits, could be used for screening grapevine cultivars for salt tolerance in oasis terroirs.

## Figures and Tables

**Figure 1 plants-12-04031-f001:**
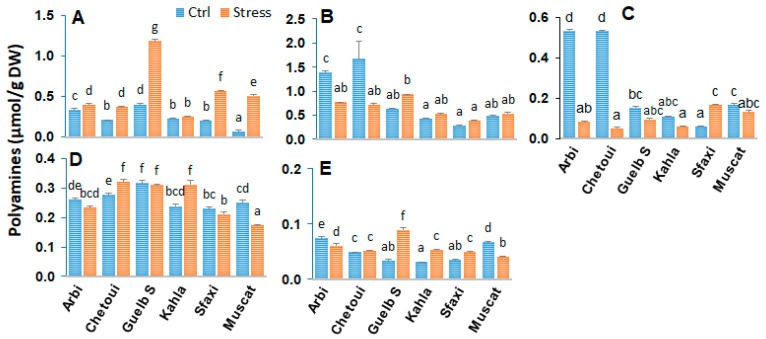
Change in the amounts of free polyamines in the berry skin of grapevine cultivars exposed to salt stress. (**A**) 1,3-diaminopropane; (**B**) putrescine; (**C**) cadaverine; (**D**) spermidine; (**E**) spermine. Data are means ± SD from three biological replicates. Different letters indicate statistically significant differences, determined via an ANOVA followed by the Newman–Keuls test, *p* < 0.05.

**Figure 2 plants-12-04031-f002:**
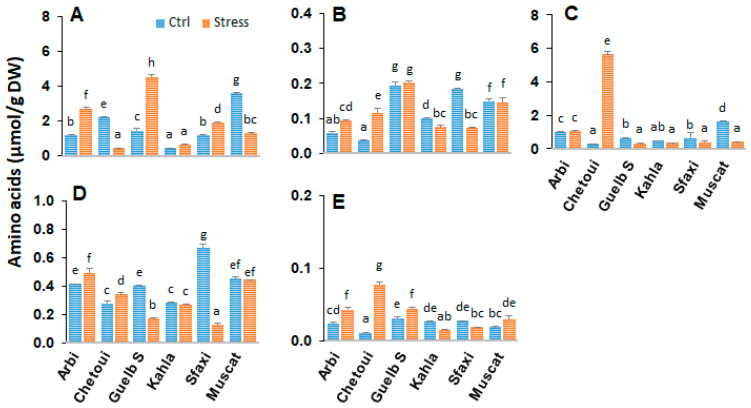
Change in the amounts of amino acids in the berry skin of grapevine cultivars exposed to salt stress. (**A**) Arginine; (**B**) Glutamate; (**C**) Proline; (**D**) γ-aminobutyric acid; (**E**) Phenylalanine. Data are means ± SD from three biological replicates. Different letters indicate statistically significant differences, determined via an ANOVA followed by the Newman–Keuls test, *p* < 0.05.

**Figure 3 plants-12-04031-f003:**
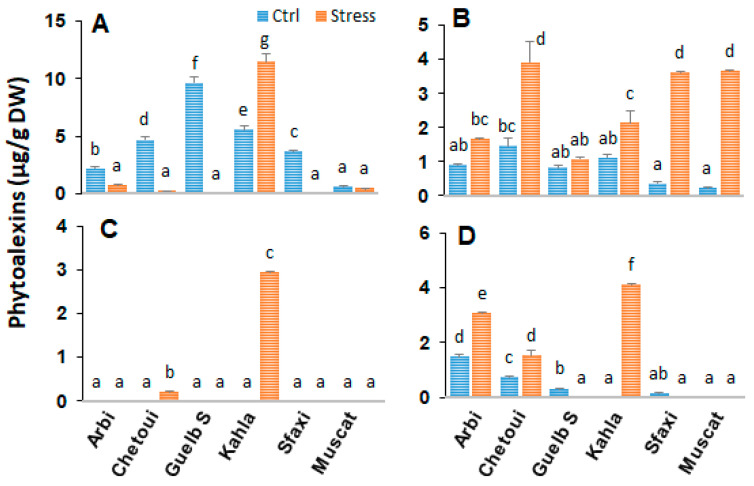
Change in the amounts of stilbene phytoalexins in the berry skin of grapevine cultivars exposed to salt stress. (**A**) Piceid; (**B**) Resveratrol; (**C**) ε-Viniferin; (**D**) δ-Viniferin. Data are means ± SD from three biological replicates. Different letters indicate statistically significant differences, determined via an ANOVA followed by the Newman–Keuls test, *p* < 0.05.

**Figure 4 plants-12-04031-f004:**
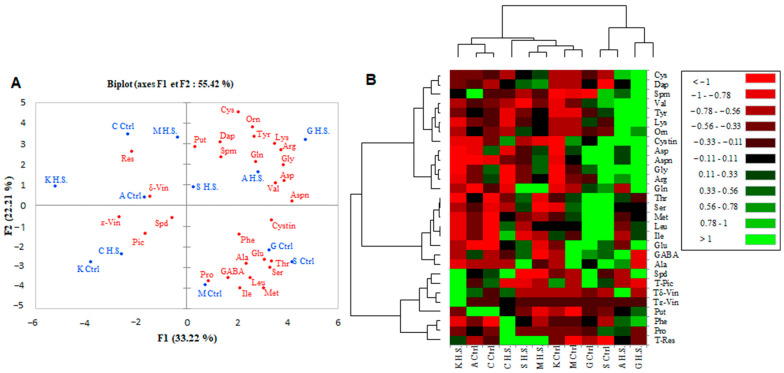
PCA plot (**A**) and heatmap (**B**) for the metabolic composition of the berry skin of oasis grape cultivars in response to salt stress. A false color scale is used to show the level of each metabolite in control and salt-stressed cultivars. Green shades indicate overproduction, and red corresponds to lower amounts of metabolites. Ctrl: control, H.S.: high salinity. The table grape cultivars: A: Arbi, C: Chetoui, G: Guelb Sardouk, K: Kahla, S: Sfaxi, and M: Muscat d’Italie.

**Figure 5 plants-12-04031-f005:**
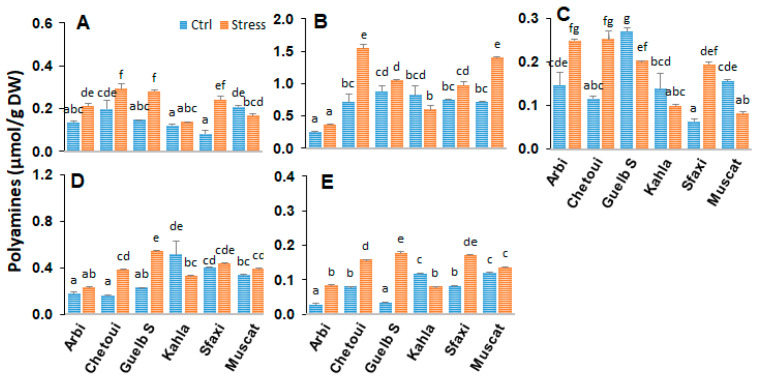
Change in the amounts of free polyamines in the berry pulp of grapevine cultivars exposed to salt stress. (**A**) 1,3-diaminopropane; (**B**) putrescine; (**C**) cadaverine; (**D**) spermidine; (**E**) spermine. Data are means ± SD from three biological replicates. Different letters indicate statistically significant differences, determined via an ANOVA followed by the Newman–Keuls test, *p* < 0.05.

**Figure 6 plants-12-04031-f006:**
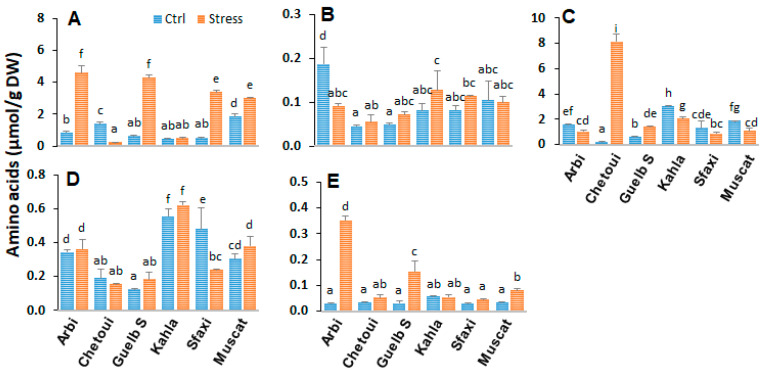
Change in the amounts of amino acids in the berry pulp of grapevine cultivars exposed to salt stress. (**A**) Arginine; (**B**) Glutamate; (**C**) Proline; (**D**) γ-aminobutyric acid; (**E**) Phenylalanine. Data are means ± SD from three biological replicates. Different letters indicate statistically significant differences based on an ANOVA followed by the Newman–Keuls test, *p* < 0.05.

**Figure 7 plants-12-04031-f007:**
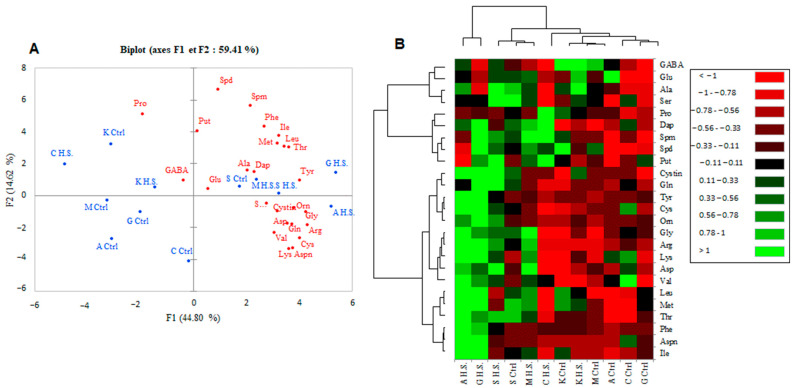
PCA plot (**A**) and heatmap (**B**) for the metabolic composition of the berry pulp of oasis grape cultivars in response to salt stress. A false color scale is used to show the level of each metabolite in the control and salt-stressed cultivars. Green shades indicate overproduction, and red corresponds to lower amounts of metabolites. Ctrl: control, H.S.: high salinity. The table grape cultivars: A: Arbi, C: Chetoui, G.S.: Guelb Sardouk, K: Kahla, S: Sfaxi, and M: Muscat d’Italie.

**Figure 8 plants-12-04031-f008:**
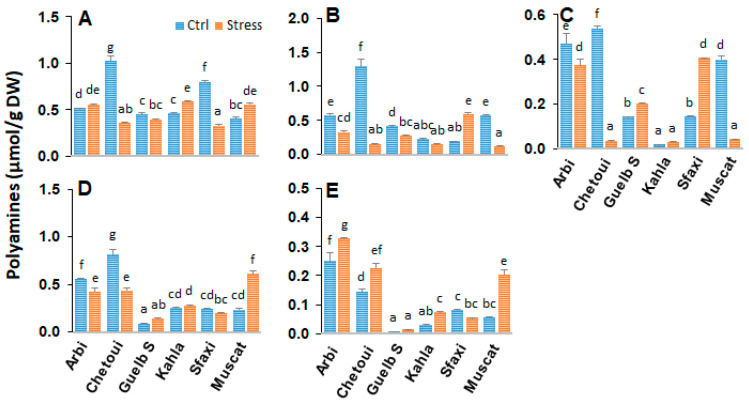
Change in the amounts of free polyamines in the berry seeds of grapevine cultivars exposed to salt stress. (**A**) 1,3-diaminopropane; (**B**) putrescine; (**C**) cadaverine; (**D**) spermidine; (**E**) spermine. Data are means ± SD from three biological replicates. Different letters indicate statistically significant differences based on an ANOVA followed by the Newman–Keuls test, *p* < 0.05.

**Figure 9 plants-12-04031-f009:**
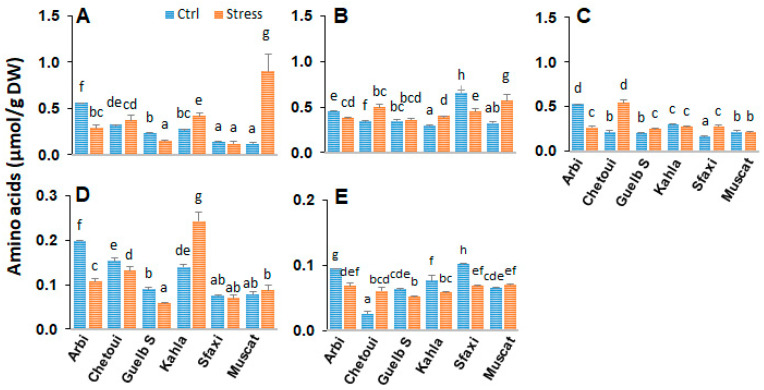
Change in the amounts of amino acids in the berry seeds of grapevine cultivars exposed to salt stress. (**A**) Arginine; (**B**) Glutamate; (**C**) Proline; (**D**) γ-aminobutyric acid; (**E**) Phenylalanine. Data are means ± SD from three biological replicates. Different letters indicate statistically significant differences based on ANOVA followed by the Newman–Keuls test, *p* < 0.05.

**Figure 10 plants-12-04031-f010:**
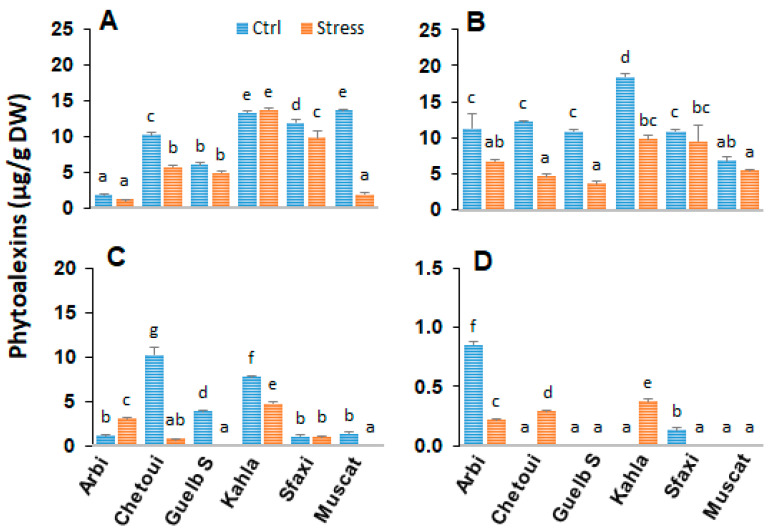
Change in the amounts of stilbene phytoalexins in the berry seeds of grapevine cultivars exposed to salt stress. (**A**) Piceid; (**B**) Resveratrol; (**C**) ε-Viniferin; (**D**) δ-Viniferin. Data are means ± SD from three biological replicates. Different letters indicate statistically significant differences based on an ANOVA followed by the Newman–Keuls test, *p* < 0.05.

**Figure 11 plants-12-04031-f011:**
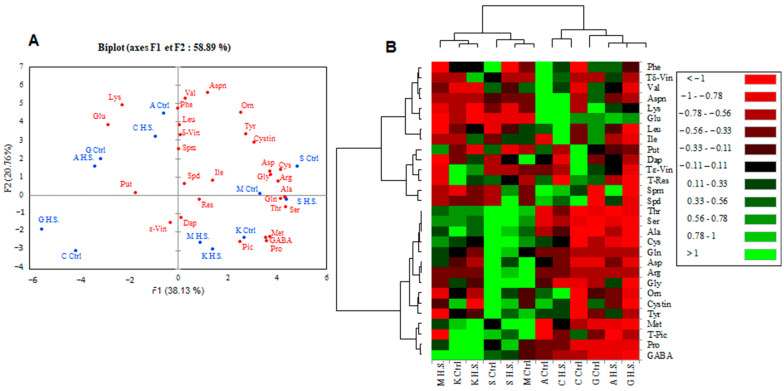
PCA plot (**A**) and heatmap (**B**) for the metabolic compositions of berry seeds of oasis grape cultivars in response to salt stress. A false color scale was used to show the level of each metabolite in the control and salt-stressed cultivars. Green shades indicate overproduction, and red corresponds to lower amounts of metabolites. Ctrl: control, H.S.: high salinity. The table grape cultivars: A: Arbi, C: Chetoui, G: Guelb Sardouk, K: Kahla, S: Sfaxi, and M: Muscat d’Italie.

## Data Availability

All data is contained within the article or Appendix A.

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
