# Peer review of "Evaluation of Salt Stress-Induced Changes in Polyamine, Amino Acid, and Phytoalexin Profiles in Mature Fruits of Grapevine Cultivars Grown in Tunisian Oases"

_plants, 2023, doi:10.3390/plants12234031_

Round 1

Reviewer 1 Report

Comments and Suggestions for Authors

This study investigated the polymines, amino acids, stilbenes and metabolic, etc., in different parts of the ripe berries of indigenous grapevine varieties after exposure to salt stress.

This study prvide the usefull clue for screening salt- tolerance grapevine.

So, i think this study is deserves to be accepted.

to identify the key traits of salt stress 18 tolerance under oasis conditions.

Author Response

Thank you for encouraging comment. Kind regards

Reviewer 2 Report

Comments and Suggestions for Authors

In this manuscript, the authors studied the salt stress-induced changes in polyamine, amino acid and phytoalexin profiles in grape berry skin, pulp and seeds of grapevine varieties cultivated in Tunisian. They observed that salinity increased amino acids strongly in the pulp and skin, while the phytoalexins as resveratrol, trans-piceid and trans-É›-viniferin, significantly accumulated in the seeds. In addition, increases in essential amino acids were also observed in both berry skin and pulp. Polyamines appeared more stable between tissues. Despite, this general behavior the authors found differences in metabolic profiles between cultivars under high salinity. Overall, the research appeared well-conducted and the manuscript was satisfactorily well written however, some points need to be corrected or clarified.

- Why the polyamine “Cad” was not included in the figures?

- Why no information on the Sfaxi cultivar was given in suppl table S1?

- Line 35. The authors state that “while both amino acids declined for the other grape varieties” however, amino acids have not changed in some varieties!

- The legend of Figure 3 is incorrect.

- line 148. It is necessary to advise that this statement is under control condition.

- These statements do not appear correct regarding the Figure  (lines 208 to 210) “Chetoui and Muscat produced high levels of amino acids in their berry pulp, while the other varieties displayed intermediate amounts.” Lines (213 to 214) “However, except for the large accumulation of proline, the high salinity drastically affected amino acid level in the berry pulp of Chetoui (negatively affected? Or not changed...).”

- Line 234. “Asp” instead “Aspn”?

- Call the Figures and table in the text when discussing the data in the discussion section.

- Please, in the discussion (lines 325 to 327) clarify the phenotypic differences associated with stress tolerance among all cultivars evaluated in this study.

Author Response

The research appeared well-conducted and the manuscript was satisfactorily well written however, some points need to be corrected or clarified.

  • Why the polyamine “Cad” was not included in the figures?

We apologize for the omission. The cadaverine was now included in the polyamine figures and discussed accordingly.

  • Why no information on the Sfaxi cultivar was given in suppl table S1?

We apologize for the inconvenience, the information on the Sfaxi cultivar was added to the suppl table S1.

  • Line 135. The authors state that “while both amino acids declined for the other grape varieties” however, amino acids have not changed in some varieties!

This was rectified according to your suggestion (Line 148).

  • The legend of Figure 3 is incorrect.

The legend was corrected (Line 176)

  • line 148. It is necessary to advise that this statement is under control condition. It was specified (Line 163)
  • These statements do not appear correct regarding the Figure (lines 208 to 210) “Chetoui and Muscat produced high levels of amino acids in their berry pulp, while the other varieties displayed intermediate amounts.” Lines (213 to 214) “However, except for the large accumulation of proline, the high salinity drastically affected amino acid level in the berry pulp of Chetoui (negatively affected? Or not changed...).” The statement was modified (L223-231)
  • Line 234. “Asp” instead “Aspn”? It was checked.
  • Call the Figures and table in the text when discussing the data in the discussion section. It was done.
  • Please, in the discussion (lines 325 to 327) clarify the phenotypic differences associated with stress tolerance among all cultivars evaluated in this study.

The relevant information was added to clarify the phenotypic characteristics of the studied cultivars (Lines 346-358).

Reviewer 3 Report

Comments and Suggestions for Authors

Dear Authors,

The manuscript received for review with the title „Evaluation of salt stress-induced changes in polyamine, amino acid, and phytoalexin profiles in mature fruits of grapevine varieties cultivated in Tunisian oases” is of interest based on the problems caused by the increase in salinity in the vineyard soils.

It is recommended to carefully check the way the manuscript was drafted so that it respects the technical editing rules of the journal, as well as the international units of measurement... as well as their correction in the manuscript. But also the detailed presentation of the analytical conditions of the equipment used to determine the monitored parameters.

Congratulations to the authors.

Author Response

It is recommended to carefully check:

-the way the manuscript was drafted so that it respects the technical editing rules of the journal, as well as the international units of measurement...as well as their correction in the manuscript. This was checked and arranged adequately.

-But also, the detailed presentation of the analytical conditions of the equipment used to determine the monitored parameters. This was arranged adequately.

Reviewer 4 Report

Comments and Suggestions for Authors

This work aims to investigate the effects of salt stress on the metabolic composition of berry skin, pulp, and seeds in various grapevine varieties cultivated in Tunisian oases. While the study provides valuable insights into the metabolic responses of these grapevine varieties to salt stress. While the research addresses an important agricultural issue (salt stress in grapevines), the paper has several critical flaws, including the lack of clear research objectives, insufficient statistical analysis, and limited contextualization

Major comments: 
1. The paper lacks a clear statement of hypotheses or research questions, making it challenging to assess the significance of the findings and their relevance to the field.

2.The paper describes the changes in metabolite levels but does not provids depths into the underlying molecular mechanisms or pathways responsible for these changes. This omission limits the paper's contribution to our understanding of salt stress responses in grapevines. 

3. The study focuses on specific grapevine varieties cultivated in Tunisian oases. It is unclear whether the findings can be generalized to other grapevine varieties or different environmental conditions. Without this context, it is challenging to assess the broader significance of the research.

Comments on the Quality of English Language

see above comments

Author Response

  • The paper lacks a clear statement of hypotheses or research questions, making it challenging to assess the significance of the findings and their relevance to the field.

The hypotheses were clearly stated in the end of the introduction (Lines 92-95, 101-106) and in the discussion sections (Lines 346-358).

  • The paper describes the changes in metabolite levels but does not provide depths into the underlying molecular mechanisms or pathways responsible for these changes. This omission limits the paper's contribution to our understanding of salt stress responses in grapevines. 

 The obtained results were adequately discussed pointing out the mechanisms and pathways involved in metabolic changes salt stress tolerance.

  • The study focuses on specific grapevine varieties cultivated in Tunisian oases. It is unclear whether the findings can be generalized to other grapevine varieties or different environmental conditions. Without this context, it is challenging to assess the broader significance of the research.

The relevant statements have been included in the different parts of the conclusions section.

Round 2

Reviewer 4 Report

Comments and Suggestions for Authors

The paper can be accepted for publication as the author have thoroughly suplemented the revised version based on my previous concerns.

Comments on the Quality of English Language

minor correction needed. 

Author Response

The manuscript was revised and the English language was edited